# Clinical Features and Physiological Signals Fusion Network for Mechanical Circulatory Support Need Prediction in Pediatric Cardiac Intensive Care Unit

Antonio Mendoza[1], Sebastian Tume[2,3], Kriti Puri[2,3], Sebastian Acosta[2] and Joseph R. Cavallaro[1]

*Abstract*—We link the hemodynamic response to inotropic agents with outcomes related to Mechanical Circulatory Support (MCS) by analyzing physiological time series and clinical features using a Machine Learning/Deep Learning ensemble approach for multi-modal waveforms in the pediatric cardiac intensive care setting of a quaternary-care hospital. Unlike existing studies that typically process a single feature type or focus on short-term diagnoses from physiological signals, our novel system processes minute-by-minute multi-sensor data to identify the need for MCS in patients admitted with Acute Decompensated Heart Failure. The data used includes tabular clinical features, time series from Intensive Care Unit monitors, and raw waveforms from electrocardiogram and arterial blood pressure signals. Our predictions facilitate early identification of high-risk patients after just two days of admission, with classification and feature importance results confirming the predictive ability of the early hemodynamic response to inotropic agent administration, achieving an AUC of 0.88 in the prediction classification task. This is particularly significant in cases where clinical decisions are not straightforward, such as those in the cohort for this study.

*Index Terms*—Blood pressure, Convolutional neural networks, Deep Learning, Ensemble learning, Hemodynamics, Heart Failure, Time series analysis, Intensive Care Unit

## I. INTRODUCTION

Heart disease is the leading cause of death in the United States [1]. As the condition worsens, patients may develop Acute Decompensated Heart Failure (ADHF), a stage where immediate medical intervention is necessary. To stabilize these patients, clinicians often use inotropic medications that help strengthen the contractions of the heart and improve the blood flow. However, some patients do not respond sufficiently to the treatment with inotropic drugs. In those cases, the patients may need a Mechanical Circulatory Support (MCS) device implanted, commonly known as a mechanical heart pump. The implantation of MCS devices carries significant risks and requires a complex surgical operation [2].

To better understand the underlying mechanics of advanced heart failure and the need for MCS implantation, we first examine what makes a patient need a MCS device implantation. A source of information, which can capture granular trends and dynamics, with rich multi-modal waveforms and time series signals, are the data streams from Intensive Care Unit

A. Mendoza and J.R. Cavallaro were supported in part by DOD grant HT9425-23-1-0663 and NIH grant 1R01HL166724-01A1.
[1] Rice University, Houston, TX, USA. Correspondence: antonio@rice.edu
[2] Baylor College of Medicine, Houston, TX, USA
[3] Texas Children's Hospital, Houston, TX, USA

(ICU) monitors. The data includes ECG, and hemodynamic parameters such as arterial blood pressure, central venous pressure and pulse oximetry. Analyzing this information can add additional insights about the status of a patient.

Patients admitted to the ICU with ADHF are in poor health, but there is no unequivocal marker or measurement to determine whether they will require MCS implantation. Tools that assess the probability of a patient recovering successfully or needing MCS implantation serve as clinical aid, enabling better planning or preparation in case there is a high chance of MCS need.

In this work we look at physiological waveforms and time series - rich and continuous information - during the first 48 hours after admission or inotropic therapy start to predict the probability of that patient needing a MCS device implantation. We analyze data of ADHF patients for a pediatric single-site Cardiac Intensive Care Unit patient cohort. We build a Machine Learning/Deep Learning system to predict whether the patient will need MCS device implantation, further deriving insights about signal and feature importance for this classification task.

Section II looks at related work. Section III details the study characteristics, and the dataset used. Section IV describes the preprocessing, architecture, and training of the models for each data type used. Section V includes the performance results and evaluation of the system. Finally, Section VI presents the conclusions of this study.

## II. RELATED WORK

While not specifically working with cardiac patients and ECG features, an XGBoost implementation of gradient boosted trees was used to classify diseases from patients with intracerebral hemorrhage (ICH) from the neurological ICU at Columbia University Medical Center [3]. That study processed features related to respiratory rate, end-tidal carbon dioxide and oxygen saturation, as well as heart rate and mean arterial pressure. ICU data has also been used in a Machine Learning scheme for detecting some arrhythmias from ECG signals. In another study [4], the authors focus on alarm signals, analyzing ECG waveform slices of up to 15 seconds. Rahman et al. have used ICU clinical, vital signs and laboratory measurements (point measurements) to predict whether hemodynamic instability was going to happen one hour in the future [5], developing a Machine Learning boost

tree based model to this end. In our previous work, ECG signals have been studied to evaluate potential candidates for Left Ventricular Assist Devices, a type of MCS device [6]. The authors only looked at ECG as snapshots in time and not a progression, and did not analyze ECG streams from the ICU setting. Yao et al. proposed a Machine Learning system for candidate identification for advanced heart failure therapies, based on a retrospective study of two patient registries with clinical data [7]. ADHF admissions of pediatric patients have been studied for risk stratification with the goal of enabling better planning [8]. However, no studies have been found that tackle specifically MCS need by taking physiological signals directly as input.

In contrast to previous related studies, which focus on one modality - often clinical features - or do not try to measure an outcome in the future, we focus on leveraging the multi-modal physiological data available from ICU monitors to make predictions on MCS implantation need by examining the hemodynamic response to inotropic agent therapy.

## III. DATA SELECTION AND PARAMETERS

### A. Study Characteristics

The objective is outcome prediction for patients admitted with ADHF. The competing outcomes included need for MCS support and successful recovery allowing for hospital discharge without planned MCS device implantation. We hypothesize that the hemodynamic response during the first hours after the start of inotropic agents administration is an indicator of recovery or worsening that will require subsequent MCS device implantation.

This is a single-center, retrospective study. The setting is pediatric Cardiac Intensive Care Unit (CICU) of Texas Children's Hospital, Houston, Texas.

*Inclusion criteria:*
- Patients admitted to the Cardiac Intensive Care Unit between 2015 and 2018.
- ICU monitor data has been recorded in the system and is retrievable.
- Blood Pressure waveform was acquired using invasive blood pressure monitoring.
- Patient was supported with inotropic agents such as Milrinone or Epinephrine.

*Exclusion criteria:*
- Patients supported with Extracorporeal Membrane Oxygenation (ECMO) therapy in the first three days of hospital stay.

Table I shows the patient demographic and clinical profile. There are no meaningful differences between the patients that needed MCS and the ones who did not. These unhealthy patients have some underlying condition that required them to be admitted to the Cardiac ICU. Furthermore, these patients needed an arterial line placement to capture their blood pressure waveforms. That means that their outcome was not easy to predict by the clinicians and required close monitoring, as invasive capture of blood pressure data with an arterial line is done only done when necessary.

TABLE I
PATIENT DEMOGRAPHICS AND CLINICAL PROFILE. VALUES ARE MEDIAN (IQR) OR NUMBER (%)

| Parameter | Patients that needed MCS (%) | Patients that did not need MCS (%) |
|---|---|---|
| Participants (n) | 50 (47.16%) | 56 (52.83%) |
| Sex (F) | 25 (50%) | 27 (48.21%) |
| Age, Years | 9.14 (12.1) | 5.68 (13) |
| Weight, Kg | 40.45 (47.8) | 18.95 (45.97) |
| Height, m | 1.28 (0.77) | 1.08 (0.91) |
| BSA, $m^2$ | 1.13 (1.06) | 0.75 (1.15) |
| SBP[1] | 97.5 (23.5) | 97.5 (31.5) |
| DBP[1] | 58 (17.25) | 58 (22.5) |
| HR[1] | 133 (40) | 133 (37.75) |
| **Left ventricular function (LVEF)** | | |
| Normal | 0 (0%) | 3 (5.35%) |
| Mildly depressed | 0 (0%) | 4 (7.14%) |
| Moderately depressed | 4 (8%) | 4 (7.14%) |
| Severely depressed | 46 (92%) | 38 (80.35%) |
| **Inotropic agent therapy received** | | |
| Milrinone | 49 (98%) | 54 (96.42%) |
| Dopamine | 7 (14%) | 9 (16.07%) |
| Dobutamine | 3 (6%) | 4 (7.14%) |
| Epinephrine | 42 (84%) | 38 (67.85%) |

[1] Taken at admission to the Unit

### B. Dataset

Following the inclusion and exclusion criteria, the final cohort is left with 106 patients. The dataset for the study includes clinical features gathered at admission (tabular data); and time series of physiological signals, acquired by electrical sensors (Electrocardiogram), pressure transducers (Arterial Blood Pressure) or Photoplethysmography (PPG, Pulse Oximeter). The signals used include low-frequency (0.5Hz) time series and high-frequency (240Hz) waveforms.

The Hemodynamic monitors used in ICU are GE Carescape B850 Version 2 monitors. Not all patients had the same signals and channels recorded. We included signals of interest that are present in all patients of the cohort in the models, for 48 hours following their first inotropic agent administration or their admission to the CICU unit, whichever is last.

*1) ECG:* The ECG monitoring was composed of leads I, II, III and V1 sampled at 240Hz. This raw data is our high-frequency data, and is also processed by the ICU monitor to calculate the low-frequency output being saved every 2 seconds. The industry-standard low-frequency time series generated by the ICU monitor are: Beats per Minute (BPM); Premature Ventricular Contraction (PVC) event alarm; ST segment deviation[1] of leads aVF, aVL, aVR, I, II, III, and V1. The signals in this ICU setting often have stronger noise than typical 12-lead ECG studies. The signals have noise that can be caused by patient movement, and can have disconnection periods during which the patient was moved for a laboratory test or for other procedures.

---

[1]ST segment refers to the interval in the ECG waveform between the S and T fiducial points of the ECG. The magnitude of this interval - the deviation - is used as a clinical marker.

*2) Arterial Blood Pressure (ABP):* The arterial blood pressure was monitored continuously using invasive arterial catheter, and any disturbance, such as nurses taking measurements or patient movement, as well as line movement, adds noise to the signal. The signal is preprocessed and filtered with a high-pass and wavelet filters to avoid distortion. In addition to the systolic and diastolic pressure values, the pressure waveform carries information about the vasculature of the patient. The high-frequency ABP waveform is used, as well as the low-frequency Systolic Blood Pressure (SBP) values and Diastolic Blood Pressure (DBP) values. The SBP and DBP magnitudes are subtracted to get the Pulse Pressure (PP) signal.

*3) Pulse Oximeter:* The patients in the cohort have PPG sensors placed to gather information about the oxygen saturation in their blood. The sensors offer complementary information about the patient's hemodynamics. The oxygen saturation (SPO2) values reported by the hospital monitors are used in this work, saved with a frequency of 0.5 Hz.

## IV. METHODS

To strive for an accurate prediction of the MCS implantation outcome, all data types explained in Section III-B are used. Hence, we have modules for clinical features, for low-frequency data, and for high-frequency data. Fig. 1 depicts a diagram of the proposed system.

### A. Preprocessing

*1) Missing Data and Outliers in Monitor Signals:* The workflow included acquiring the signals from the database, performing file format conversion, exploring signals present, and evaluating if the signal actually contained a valid waveform (and not mostly noise, zero, infinite or Not-a-Number values).

For this study, missing data and outlier instances are considered invalid samples. We detect outliers in the low-frequency data sample-by-sample based on thresholds, and their value in the time series was imputed according to criteria explained in this section. Outliers such as a BPM of 500 or SBP of -20 for a few samples were present, making this detection and imputation process necessary. The thresholds set for the signals are shown in Table II.

*2) Minimizing age-dependency in the low-frequency features:* Adjustments to some features were made to minimize age dependency. First, Pulse Pressure is used, defined as the difference between the SBP and the DBP measurements, thus eliminating the pressure absolute values. Second, heart rate related features that use the absolute value (i.e. mean, max, min) are post-processed by replacing the absolute values with the Z-score based in the age range. The values used belong to a systematic review of observational studies reported in [9]. When estimating, the authors used Pearson's 2nd coefficient and Bowley skewness tests and observed no skewness in the heart rate data, hence assumed a normal distribution at each age. We will also assume a normal distribution at each age for

TABLE II
THRESHOLDS AND SAMPLE IMPUTATION TECHNIQUES FOR THE LOW-FREQUENCY (0.5HZ) SIGNALS OF THE PATIENTS IN THE STUDY COHORT.

| Signal | Min. Threshold | Max. Threshold | Imputation Technique |
|---|---|---|---|
| Beats per Minute | 30 | 220 | Last valid value |
| PVC alarm | 0 | 30 | Replace with 0 |
| ST Segment deviation | -4 | 4 | Last valid value |
| SPO2 | 60 | 100 | Replace with 100 |
| Pulse Pressure | 10 | 70 | Last valid value |

this study. The age of each patient to perform the appropriate adjustment was obtained from the clinical data.

Two assumptions were made when defining the strategy for imputation: 1) During the missing or invalid sample, assume there is no new alarm (assume normalcy), and 2) During the missing or invalid sample, there were no major changes in the physiological process (assume short-term stability). In our dataset, most of the invalid samples are short gaps between valid values, making the two assumptions reasonable. Hence, when we need to impute a sample, we use the non-alarm values for the case of PVC and SPO2 signals, or we use the Last-observation-carried-forward (LOCF) approach for BPM, PP and ST Segment deviation samples. In the case where the first sample of a slice is invalid, we impute its value with the mean of all valid samples in the slice. When more than half of the samples in a 5-minute segment were marked as invalid, that 5-minute segment slice is marked as invalid at the segment level. If a 5-minute segment is marked as invalid, their statistical values are imputed, replacing them with the statistical values of the last valid 5-minute segment, following the assumptions and the LOCF approach. Lastly, when a patient signal abruptly ends (end of recording) before the full study period (as is the case in this dataset for some ABP instances), the remaining 5-minute slices are imputed with the average values of the last 12 slices (representing the last hour with valid data).

*3) Filtering:* High-frequency signals - ECG and ABP waveforms - were filtered. To remove the baseline wander, a 4th order high-pass Butterworth filter, with a cutoff frequency of 0.5 Hz is used, applying the filter twice to achieve zero phase distortion, using the scipy.signal Python package. To minimize other noise, wavelet based filtering using the Symlets 4 (sym4) wavelet with a high level of decomposition is used after baseline wander removal. Wavelet-based denoising is a commonly used algorithm for noise removal in ECG signals [10]. The threshold that defines noise is set experimentally to 0.04. Coefficients below the threshold are set to zero to eliminate the noise when doing the reconstruction. Skewness and kurtosis are calculated and segments with outlier values are marked as invalid slices.

All models were evaluated by performing K-Fold Cross-validation, recommended for datasets with limited data, as is the present study. We are using stratified K-fold, to ensure that

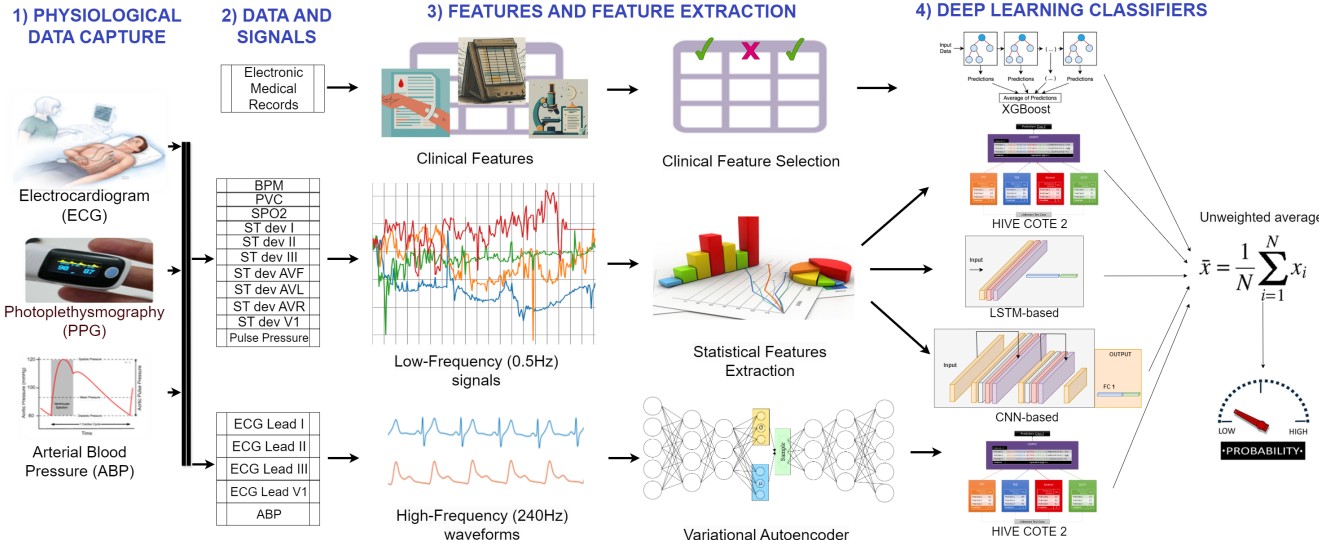

Fig. 1. Proposed architecture for the system for MCS outcome prediction. From left to right: 1) The physiological signals (ECG, PPG and ABP) that are captured in the hospital system from the ICU monitors. 2) The data (clinical features extracted from the patient records) and the signals, comprised of low-frequency (0.5Hz) measurements from the ICU monitor and raw high-frequency waveforms. 3) The features used as input for the classifiers and the methods used for feature extraction. 4) The Deep Learning classifiers that predict the probability, members of the ensemble. Finally, an unweighted average is done to obtain the final predicted probability.

each fold keeps the same proportion of negative and positive classes. The value of K chosen was 5, as the cohort is limited, and 5-fold is generally recommended as a good compromise between variance and bias [11]. Results presented are obtained from 5-fold cross-validation. A Repeated Stratified K-Fold Cross-validation scheme is used, with 10 repeats, as repeating the K-fold cross-validation can be used to effectively increase the precision of the estimates while still maintaining a small bias [12]. Each repeat is performed with a different fold split, ensuring that each time an observation is part of the test subset inside a fold, the training subset contains different training observations. The Scikit-learn Python package and its implementation in the *RepeatedStratifiedKFold* class are used in this work. Following this scheme, each observation is predicted 10 times, one per repeat of the 5-fold cross-validation process, when the observation is part of the test subset.

The final predicted probability is computed with an ensemble of the models used, as ensembles are known to provide better performance than any single learner [13]. A fusion strategy using unweighted model averaging has been employed, avoiding weighted averaging to prevent bias. In Section V, all the results of the modules and of the system are shown. This Section details the modules used, with their processing steps, input data and architectures.

### B. Clinical Features Module

The first module consists of a model that uses only clinical features as input for the classification task. To this end, we trained a XGBoost model [14] on the clinical features available from the dataset, and then performed feature selection, leaving a XGBoost classifier with 17 features: Age, sex, pre-existing heart failure, previous admission or not, Body Surface Area (BSA), Left Ventricular Ejection Fraction (LVEF), shock at admission or not, heart rate, if the patient had previous Ventricular Tachycardia episodes, and levels of BNP [2], BUN [3], sodium, creatinine, total bilirubin, hematocrit, potassium and hemoglobin.

To select the clinical features, first, a preliminary selection using feature importance reports from earlier runs of XGBoost models that started with all features was used. Secondly, features chosen had to be available at admission time, excluding other lab tests conducted after admission to the ICU, or diagnosis-related features. Lastly, features usually associated with heart function for monitoring for heart failure, and used for predicting heart failure re-admissions in the literature, were considered [15]. A grid search method was used to obtain the best hyperparameters, achieving best results with $\gamma = 0.2$, $maxTree = 7$, $subsampleRatio = 0.8$ and $subsampleRatioColumns = 0.8$. We are using the default trees as base predictors [14].

### C. Low-frequency module

This module processes the low-frequency signals. We preprocess these signals as described in Section IV-A, then extract features as explained below, and feed them to the Deep Learning model.

[2]Natriuretic peptide tests measure levels of B-type natriuretic peptide (BNP) and N-terminal pro-B-type natriuretic peptide (NT-proBNP) hormones, released by the heart. Elevated levels of these peptides are indicative of heart failure and other cardiac conditions.

[3]Blood Urea Nitrogen (BUN) tests can evaluate renal function and metabolic health. Elevated BUN levels may signify impaired renal function, dehydration, or cardiac-related issues.

To reduce the dimensionality, we extracted statistical features from the time series data. The two-day time series signal is sliced into non-overlapping 5-minute windows. Slices of 5 minutes were chosen, as they are frequently used in the Heart Rate Variability [16] field, since they capture enough variability to be useful in analysis. For our purposes, these 5-minute segments capture enough data to be useful as representation for the Deep Learning models. We extract the statistical features out of every 5-minute window, and these features are the inpuy of the downstream classification task.

The statistical values obtained include mean, min, max, standard deviation of the variable values in the slice, as well as of the difference array (comprised of the results of the difference between consecutive variable values $x_{i+1} - x_i$). The root mean square of successive differences $rmssd$ and the coefficient of variation $cv$ are also calculated. Not all statistical features for all signals are used, only the ones shown in Table III are included in the feature array fed to the Deep Learning models. The final sets of signals and statistical features used were chosen after an ablation study that checked the sensitivity of the models to the different configurations.

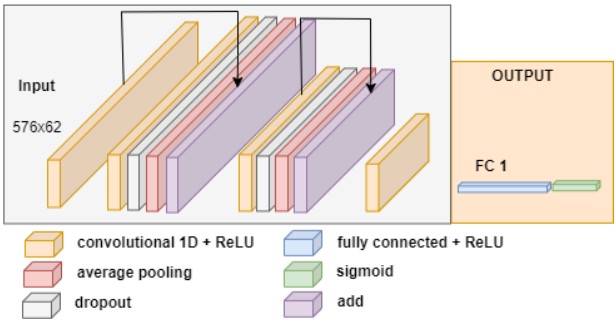

Fig. 2. Architecture of the CNN based model used with the low-frequency features. The input is 576 elements in the time series (one per 5-minute slice in the 2-day observation window), each with 62 values (the statistical features from Table III).

TABLE III
ROW INDEX IN THE PYTHON NUMPY FEATURE ARRAY OF EACH
LOW-FREQUENCY SIGNAL AND ITS STATISTICAL FEATURES.

| Signal | Mean | Standard Dev. | Min. Value | Max. Value | Range of Diff. Array, RMSSD, CV, and Sample Entropy |
|---|---|---|---|---|---|
| Heart Rate | 0 | 1 | 2 | 3 | 4-10 |
| PVC Alarm[a] | 11 | | | | |
| SPO2 | 12 | 13 | 14 | 15 | 16-22 |
| ST Segment Dev. AVF | 23 | 24 | 25 | 26 | |
| ST Segment Dev. AVL | 27 | 28 | 29 | 30 | |
| ST Segment Dev. AVR | 31 | 32 | 33 | 34 | |
| ST Segment Dev. SEG1 | 35 | 36 | 37 | 38 | |
| ST Segment Dev. SEG2 | 39 | 40 | 41 | 42 | |
| ST Segment Dev. SEG3 | 43 | 44 | 45 | 46 | |
| ST Segment Dev. V1 | 47 | 48 | 49 | 50 | |
| Pulse Pressure | 51 | 52 | 53 | 54 | 55-61 |

[a]Flag set to 1 if >3 PVC alarms trigger in the 5-minute segment.

For this low-frequency module, after evaluation of different architectures and hyperparameters, three models have been trained.

*a) CNN Residual based model:* - The first model is a convolutional based model. This model was chosen because of the success in time series classification with 1-D convolutions, which help learn the temporal dependencies and trends. Skip connections are used to allow the model to learn both long-term and short term trends.

The model is based on 1-D CNN layers, with dropout, average pooling and skip connections. It is a residual architecture, with a hyperparameter search performed using Keras Tuner, obtaining the number of filters of the convolutional layers - 32, 32, 48 and 48, respectively, all with filter size 5. The number of units in the Dense layer is 96, and the dropout rate is 0.25. Fig.

2 depicts the model layers. The model has been trained with an initial learning rate of 0.0003 using the Adam optimizer, and using Early Stopping to finish training when the monitored validation accuracy does not improve anymore after 25 epochs.

*b) LSTM based model:* - The second model is based on Long Short-Term Memory (LSTM), chosen due to its capability of keeping or discarding temporally separated information enabling the model to learn different aspects of the time series. The LSTM model used has three layers, with 192, 112, and 120 units, followed by a fully connected layer and a final sigmoid activation unit. The hyperparameters were chosen with Keras Tuner.

*c) HIVE-COTE V2 model:* - The third model trained is based on Hierarchical Vote Collective of Transformation-based Ensembles (HIVE-COTE 2). HIVE-COTE 2 is designed for time series classification, combining classifiers from different representations to improve predictive performance. The ensemble members include algorithms that use bag-of-words, shapelets and random transforms of convolutional kernels. This model is implemented and trained using the Python package Sktime.

Hyperparameters for the CNN and LSTM models were selected using Keras Tuner with two days of low-frequency data. Training instability and high variability in validation results found in the first experiments, indicated sensitivity to weight initialization and demanded additional measures. A Cosine Annealing learning rate [17] schedule was implemented, which adjusts the learning rate cyclically to explore different parts of the weight space, potentially reaching better local minima, and mitigating validation variability.

*D. High-frequency module*

High frequency ECG and ABP signals were recorded at 240 Hz as mentioned in Section III-B. To reduce the dimensionality of the data for time series classification, a $\beta$-Variational Autoencoder (VAE) scheme has been adopted together with slicing all signals to segments of 10 seconds each. This interval length is consistent with the window traditionally used for standard 12-lead ECG tests.

The architecture of this high-frequency module consists of a VAE followed by a classifier. The VAE is based on residual

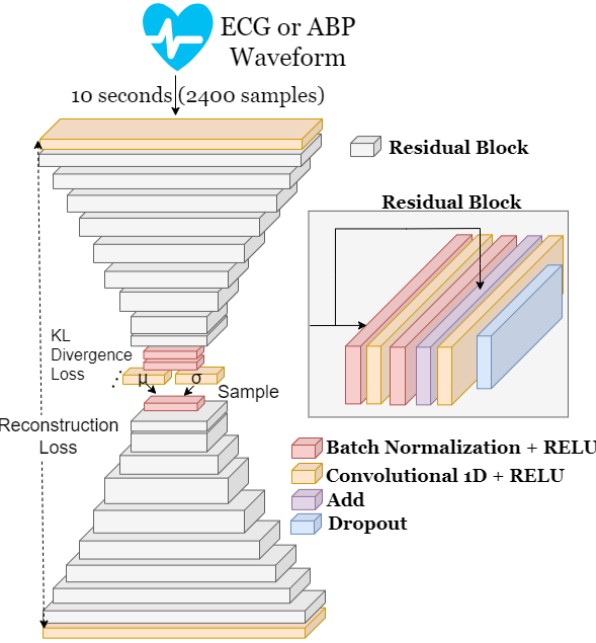

Fig. 3. Architecture of the Variational Autoencoder model used for feature extraction and dimensionality reduction of the high-frequency ECG and ABP signals.

blocks that use 1D convolutional layers. It follows typical VAE architecture patterns, with an hourglass shape in the layers. The number of filters in the encoder, decreases every 4 layers following the pattern 80, 64, 48, 32 and 16, and filter sizes are 9 and 19 in the first and second half of convolutional layers. Dropout rate is 0.1. The second convolutional layer in each residual block except the last two blocks has strides equal to 2. The mean and variance vectors have size 60. The decoder has the same structure as the encoder but without the skip connections and inverted, having as output of the model an array of length 2400 (10 s, 240 Hz). Fig. 3 illustrates the model architecture. Gridsearch was used to select hyperparameters for the layers and dropout rates. Several values of the regularization parameter $\beta$ for binary cross-entropy and KL divergence losses were tested. Lower total loss and a good reconstruction performance was achieved with $\beta = 0.005$.

After training the VAE model, the whole dataset is passed through the encoder, and we take the mean latent vector to use for the classification task. We have two days of high-frequency data available per patient, hence every patient has 17280 slices of 10 seconds each. For the classification task, we feed the time series to HIVE COTE V2, having as inputs the time series of 60 feature vectors per high-frequency signal (300 features in total) and 17280 steps per patient. The model was trained with batch size 384, using the Adam optimizer with learning rate 0.0002 and Early Stopping monitoring the reconstruction loss, ending the training when the loss did not improve for 8 epochs. As with the other modules, this module was evaluated using Repeated Stratified K-fold Cross-Validation [11] [12], with a different split in each of the 10 repeats and getting ensemble predictions for each observation

[13], using unweighted averaging.

## V. RESULTS AND DISCUSSION

We have three types of input available: Clinical features, low-frequency (0.5Hz) time series, and high-frequency (240Hz) waveforms. Five models were developed for the three types of input data, as described in Section IV:

- Input: Tabular clinical features. XGBoost model. (CF)
- Input: Low-frequency statistical features. CNN-based model (LF-CNN)
- Input: Low-frequency statistical features. LSTM-based model (LF-LSTM)
- Input: Low-frequency statistical features. HIVE-COTE V2 based model (LF-HC2)
- Input: High-frequency features, consisting of the latent space features from the Variational Autoencoder that received the high-frequency waveforms. HIVE-COTE V2 based model (HF-HC2)

Metrics computed were accuracy, precision, recall, F1 scores, ROC curves and AUC scores. Additionally, the $F_\beta$ score has been calculated. This score is mathematically expressed as shown in Equation 1.

$$F_\beta = (1 + \beta^2) \cdot \frac{\text{precision} \cdot \text{recall}}{(\beta^2 \cdot \text{precision}) + \text{recall}} \qquad (1)$$

This equation represents the $F_\beta$ score, which is a weighted harmonic mean of precision ($\frac{true\,positive\,predictions}{all\,positive\,predictions}$) and recall ($\frac{true\,positive\,predictions}{actual\,positives}$). The $\beta$ parameter controls the relative importance of precision and recall, with $\beta > 1$ emphasizing recall over precision and $\beta < 1$ emphasizing precision over recall. After expert consultation and review of the literature, we have seen that in this context of heart failure patients, minimizing false positives is preferred [18], hence we adopt $\beta = 0.2$ for this metric.

The predicted probabilities obtained by each type of model, represented as vectors with a size of 106 (patients in the cohort), where each element falls within the range of 0 to 1, have been combined with one or more other models to form ensembles. A straightforward, unweighted averaging approach has been adopted for the predicted probabilities of each patient. This approach avoids adding bias during the ensemble creation process.

We have tried 20 different configurations, summarizing their performance results in Table IV. While in general the performance is improved using ensembles (rows 4-19), the combination of clinical features, low-frequency CNN architecture, and low-frequency LSTM architecture stands out. When adding to that combination the high-frequency HC-2 classifier, we get the best faring model in all performance metrics except the AUC score. Combinations that include clinical features and at least one of the CNN-based or LSTM-based models get the best performance, which gets improved further when including both. Rows 15,16 and 19 exhibit this improvement, achieving $F_\beta > 0.85$. Those combinations also outperform models that only use the low- or high-frequency physiological data, suggesting that there is complementary information between the

clinical features and the physiological signals. In addition to that, models that use both CNN-based and LSTM-based low-frequency data outperform models that use only one of them, suggesting that these two architectures extract complementary information and benefit from the ensemble approach.

TABLE IV
PERFORMANCE RESULTS OF THE MODELS TRIED. CF: CLINICAL FEATURES, LF-CNN: LOW-FREQUENCY CNN BASED, LF-LSTM: LOW-FREQUENCY LSTM BASED, LF-HC2: LOW-FREQUENCY HIVE COTE V2, HF-HC2: HIGH-FREQUENCY HIVE-COTE V2. FIRST, SECOND, AND THIRD BEST SCORES ARE SHOWN IN BOLD, UNDERLINED AND ITALICS, RESPECTIVELY. ROWS 0-4 ARE SINGLE MODELS, 5-19 ARE ENSEMBLES.

| # | Model | Accuracy | Precision | Recall | F1 | Fbeta | AUC-score |
|---|---|---|---|---|---|---|---|
| 0 | CF | 0.736 | 0.729 | 0.7 | 0.714 | 0.728 | 0.774 |
| 1 | LF-CNN | 0.736 | 0.762 | 0.64 | 0.696 | 0.756 | 0.784 |
| 2 | LF-LSTM | 0.736 | 0.739 | 0.68 | 0.708 | 0.737 | 0.797 |
| 3 | LF-HC2 | 0.623 | 0.619 | 0.52 | 0.565 | 0.615 | 0.689 |
| 4 | HF-HC2 | 0.623 | 0.625 | 0.5 | 0.556 | 0.619 | 0.682 |
| 5 | CF, LF-CNN | 0.764 | 0.821 | 0.64 | 0.719 | 0.812 | 0.864 |
| 6 | CF, LF-LSTM | 0.764 | 0.766 | **0.72** | 0.742 | 0.764 | 0.841 |
| 7 | CF, LF-HC2 | 0.745 | 0.767 | 0.66 | 0.71 | 0.763 | 0.82 |
| 8 | CF, HF-HC2 | 0.774 | 0.783 | **0.72** | *0.75* | 0.78 | 0.793 |
| 9 | LF-CNN, LF-LSTM | 0.726 | 0.769 | 0.6 | 0.674 | 0.761 | 0.826 |
| 10 | LF-CNN, LF-LSTM, LF-HC2 | 0.745 | 0.871 | 0.54 | 0.667 | *0.851* | 0.826 |
| 11 | LF-CNN, HF-HC2 | 0.717 | 0.763 | 0.58 | 0.659 | 0.754 | 0.808 |
| 12 | LF-LSTM, HF-HC2 | 0.689 | 0.743 | 0.52 | 0.612 | 0.731 | 0.788 |
| 13 | LF-HC2, HF-HC2 | 0.679 | 0.735 | 0.5 | 0.595 | 0.722 | 0.729 |
| 14 | CF, HF-HC2 | 0.774 | 0.783 | **0.72** | *0.75* | 0.78 | 0.793 |
| 15 | CF, LF-CNN, LF-LSTM | 0.811 | 0.857 | **0.72** | 0.783 | *0.851* | 0.886 |
| **16** | CF, LF-CNN, LF-LSTM, HF-HC2 | **0.821** | **0.878** | **0.72** | **0.791** | **0.871** | *0.885* |
| 17 | LF-CNN, LF-LSTM, HF-HC2 | 0.764 | 0.821 | 0.64 | 0.719 | 0.812 | 0.835 |
| 18 | LF-CNN, LF-LSTM, LF-HC2, HF-HC2 | 0.755 | 0.833 | 0.6 | 0.698 | 0.821 | 0.834 |
| 19 | CF, LF-CNN, LF-LSTM, LF-HC2, HF-HC2 | *0.783* | *0.865* | 0.64 | 0.736 | 0.853 | **0.889** |

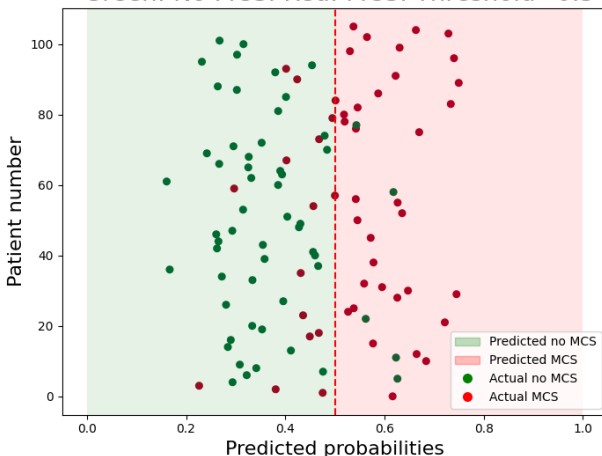

Fig. 4. Predicted probabilities of the ensemble 16 from Table IV. Each of the 106 dots is a patient prediction for this cohort. The color of the dot is the ground truth (MCS need or not), and the colored regions delimited by the vertical threshold line shown the predicted regions. Only 5 false positives can be seen (green points in the red shaded region), highlighting the high precision of the system.

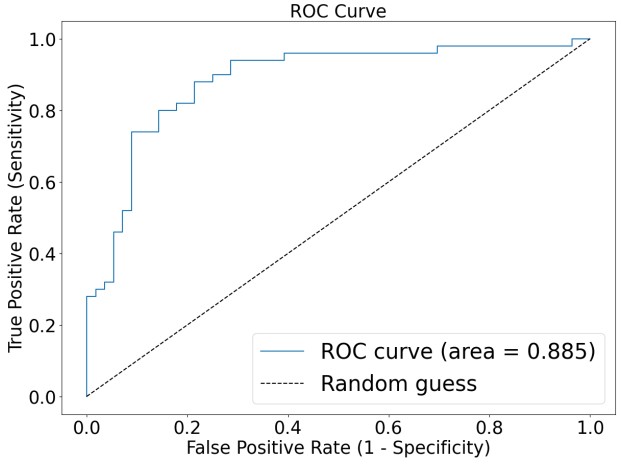

Fig. 5. ROC curve and AUC score of ensemble 16, the best performing ensemble.

Fig. 4 displays the predicted probabilities of the system with the ensemble with the highest performance in most metrics (ensemble 16), comprised of the CF, LF-CNN, LF-LSTM and HF-HC2 models. The image shows the overall accuracy and the low amount of false positives. Fig. 5 shows the ensemble's ROC and AUC scores.

This ensemble approach surpasses the expected results of AUC>0.8 to be considered as an acceptable tool in the healthcare context [19]. High precision and $F_\beta$ scores show that the system is very effective at minimizing false positives, giving confidence that when the system predicts the need for MCS implantation, clinicians can consider this confidently and plan for this possibility. This proactive planning increases the likelihood of a successful surgery and recovery.

These results highlight the potential of the system as a tool to predict the need for MCS implantation, serving as clinical aid early after admission - requiring only the first 48 hours of physiological signals. The significant improvement in classifier performance scores between the clinical features only model and models that add the time series data from the physiological signals, validates our hypothesis that hemodynamic response to inotropic agent initiation helps significantly in the prediction of successful recovery or worsening of a patient and is a valuable source of information.

This study's pediatric cohort posed challenges due to the diverse age range. Although they were addressed, there are potentially confounding factors introduced affecting generalizability. The study was constrained by the availability of a limited patient cohort and from a single site, which could

potentially affect the robustness and reliability of the results. The size of the dataset precluded the implementation of more advanced techniques, such as model stacking. Extending the dataset with larger and multi-site cohorts would potentially enhance the robustness of the classifier. In this initial study we used high-quality ABP waveforms, which are generally less available due to their invasive nature. We could expand the cohort by including additional patients with non-invasive cuff measurements. Tests would be needed to understand the sensitivity of the system and its performance with the transition from continuous waveforms to less frequent and point-wise blood pressure measurements. While we used a single-site pediatric dataset for this study, the proposed methodology is not unique to our dataset and is applicable to other multimodal datasets.

## VI. CONCLUSION AND FUTURE WORK

We present a novel multimodal framework for predicting MCS-related outcomes in ICU patients with acute decompensated heart failure by analyzing patient data from only the first 48 hours after admission. The system uses unsupervised methods for dimensionality reduction combined with statistical feature engineering to capture morphological and temporal features for supervised classification along with clinical feature input in an ensemble approach.

Our findings suggest that there is a link between the hemodynamic response to inotropic agents and patient recovery or worsening with the consequent need for MCS implantation or not. By analyzing that hemodynamic response with deep learning approaches we achieved an AUC score of 0.88 with 0.91 specificity. The complementary information from clinical features and physiological signal time series enhances model performance, as demonstrated by improved predictions when combining these inputs. To the best of our knowledge this study is pioneering in establishing this specific link, paving the way for future research in predicting successful recovery from acute decompensated heart failure.

Clinical and physiological signals offer complementary information, with the best-performing ensemble model including models from clinical, low-frequency, and high-frequency derived features. Moreover, complementary neural network architectures benefit from ensemble approaches. CNN and LSTM models, despite using the same data, weighted features differently, further supporting ensemble strategies.

The ICU outcome framework developed in the study was applied to pediatric patients. In future work, we plan to extend the study to encompass adult patients. Also, extraction of more complex features from the low-frequency data that may highlight other temporal dynamics will be explored, as well as more advanced imputing methods [20] tailoring them to our physiological signals of interest.

The current study based outcome prediction on the first two days after inotropic agent administration, as the hypothesis was that successful patient response occurs during that window of time. We can extend our framework to incorporate a sliding window approach, to have near real-time updating of predicted probabilities, allowing for dynamic risk assessment and timely intervention based on evolving patient trajectories.

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
