# OpenReview forum: "Clinical Features and Physiological Signals Fusion Network for Mechanical Circulatory Support Need Prediction in Pediatric Cardiac ICU"
_IEEE.org/EMBS/BHI/2024/Conference — IEEE BHI'24_

### Official Review · Reviewer_KJgE · 2024-07-30

**Overall Rating:** 7
**Confidence:** 5

**Other Quality Metrics:**

Clarity of writing: Fair

Clinical Significance: Excellent

Methodological Novelty: Great

Experiments and Results: Great

**Questions For The Authors:**

1)	How did you choose the methods for imputing missing values? Advanced methods for imputing missing values exist. You could mention the study in [1] as future work.
[1] Jarrett, D., Cebere, B. C., Liu, T., Curth, A., & van der Schaar, M. (2022, June). Hyperimpute: Generalized iterative imputation with automatic model selection. In International Conference on Machine Learning (pp. 9916-9937). PMLR.
2)	I assume that Table IV includes an ablation study. However, it is not obvious in this current form. Please add an “Ablation Study” section.

**Strengths:**

1)	Interesting study.
2)	A lot of experiments conducted.

**Summary Of The Paper:**

This paper describes a machine learning approach for predicting Mechanical Circulatory Support (MCS)-related outcomes in Intensive Care Unit (ICU) patients.

**Weaknesses:**

1)	The paper is not well organized. For instance, Section III is named “Methods”. It is expected that the reader will get an overview of the proposed methodology. However, the methodology is described in Sections IV – A, B, C. Please, reorganize the paper.
2)	Simple methods for imputing missing values.
3)	Is there any prior work on this topic? Have other studies experimented with predicting this task?
4)	Please cite more recent works if possible.
5)	Table IV is large and needs more elaboration.
6)	In Abstract, the limitations of existing studies are not discussed.

---

### Official Review · Reviewer_6hKb · 2024-08-09
**Revision of 182 paper**

**Overall Rating:** 7
**Confidence:** 4

**Other Quality Metrics:**

- Clarity of writing: fair.
- Clinical significance: great.
- Methodological novelty: fair.
- Experiments and results: good.

**Questions For The Authors:**

- please, avoid the use of undefined acronyms within the title.
- in Table IV each model has been labeled as ensemble, but the first four cannot be indicated as ensemble, because they are based on a single classification model, fed by a single type of signals.
- for the low-frequency signals, very simple features have been extracted. Extracting a different kind of features, that possibly highlights temporal dynamics within the data may be beneficial. This point could be discussed.
- why for the low-frequency data, three models have been considered (CNN, LSTM, and HIVE-COTE) whereas for high frequency data only one?

**Strengths:**

- the work has a significant clinical importance, since a reliable identification of the need for MCS is fundamental.
- different kind of information has been pooled together, highlighting the importance of considering multiple sources of information for this kind of problem.

**Summary Of The Paper:**

In this work an ensemble model has been proposed for predicting the need for mechanical circulatory support in pediatric patients. To this purpose, 48 hours of different signals have been leveraged, together with different classifiers. Outcomes showed that the best configuration is the one where all the information is used and combined by the ensemble of single classifiers.

**Weaknesses:**

- a relatively small cohort of patients has been considered (106).

---

### Official Review · Reviewer_41kq · 2024-08-10
**Technical deatils in this paper is promising. Howeer, findings of this method needs to be evaluated with larger datasets available on Physionet databases. Abstract needs major revision as it is too brief withut providing required details to convicne reader to read the full manuscript.**

**Overall Rating:** 7
**Confidence:** 5

**Other Quality Metrics:**

a) good
b) great
c) good
d) good

**Questions For The Authors:**

Why did you use only 10 seconds for ECG and ABP waveforms?
Did you use a signal quality index for waveforms?
Could you please comment on integrating the ensemble proposed method into the clinical workflow?

**Strengths:**

Comparison between models developed on tabular clinical features, low-frequency statistical features, and high-frequency waveforms is interesting.

**Summary Of The Paper:**

The research paper investigates the prediction of Mechanical Circulatory Support (MCS) needs in pediatric cardiac patients using an ensemble approach that combines physiological time series and clinical features. The study develops five models based on different types of input data, including clinical features, low-frequency time series, and high-frequency waveforms. Evaluation metrics. The study employs K-Fold Cross-validation with 5 folds and 10 repeats to split the data for evaluation. The results demonstrate the effectiveness of the proposed approach in predicting the need for planned MCS implantation. The study utilizes a sample size of 106 patients, and while the paper lacks an explicit discussion of limitations, it provides valuable insights into the identification of high-risk patients in the pediatric cardiac intensive care setting.

**Weaknesses:**

It is not clear how the proposed methodology is superior to previous research listed in section II. Considering the sample size and complexity of the method utilized, it is not clear why other datasets available on Physionet are not used in this study.

---

### Decision · Program_Chairs · 2024-09-23

Accept